# Reference Values and Repeatability of Pulsed Wave Doppler Echocardiography Parameters in Normal Donkeys

**DOI:** 10.3390/ani12172296

**Published:** 2022-09-05

**Authors:** Mohamed Marzok, Adel I. Almubarak, Zakriya Al Mohamad, Mohamed Salem, Alshimaa M. Farag, Hussam M. Ibrahim, Maged R. El-Ashker, Sabry El-khodery

**Affiliations:** 1Department of Clinical Scienses, College of Veterinary Medicine, King Faisal University, Al-Ahsa 31982, Saudi Arabia; 2Department of Surgery, Faculty of Veterinary Medicine, Kafrelsheikh University, Kafrelsheikh 33516, Egypt; 3Department of Internal Medicine, Infectious and Fish Diseases, Faculty of Veterinary Medicine, Mansoura University, Manosura 35516, Egypt

**Keywords:** pulsed wave, echocardiography, donkeys, reference values

## Abstract

**Simple Summary:**

Cardiovascular disease is underreported in donkeys, possibly related to their limited athletic posture and frequent poor performance-related examinations. Reports on treatments for cardiovascular disease are anecdotal in donkey. Normal echocardiographic parameters have been reported in healthy donkeys. The aim of the present study was to establish the reference values and repeatability for Pulsed Wave Doppler echocardiographic variables of the mitral valve, aortic valve and myocardial performance. Two-dimensional Color flow mapping and spectral Doppler modes were performed. For the mitral valve, the mean velocity, pressure gradient and duration of E-wave were 57.7 ± 12.5 cm/s, 1.4 ± 0.7 mmHg and 0.4 ± 0.13 s, respectively. The results of the present study provide the reference values of PW echocardiographic parameters measurements in normal adult donkeys. Such reference values are helpful, especially when confronted with clinical cases with cardiovascular disorders.

**Abstract:**

In the present study, thirty clinically healthy donkeys were used to establish the reference values and repeatability for Pulsed Wave Doppler echocardiographic variables of the mitral valve, aortic valve and myocardial performance. 2-dimensional Color flow mapping and spectral Doppler modes were performed. For the mitral valve, the mean velocity, pressure gradient and duration of E-wave were 57.7 ± 12.5 cm/s, 1.4 ± 0.7 mmHg and 0.4 ± 0.13 s, respectively. The velocity, pressure gradient and duration of the A-wave were 32.3 ± 9.1 cm/s, 0.3 ± 0.04 mmHg and 0.3 ± 0.1 s, respectively. The mitral valve area, pressure half time, pulsatility index (PI), resistance index (RI) and velocity time integral (VTI) were 1.8 ± 0.5 cm^2^, 66 ± 17 ms, 2.8 ± 1.4, 0.9 ± 0.03 and 19.1 ± 5.7 cm, respectively. For the aortic valve, the mean velocity was 64.9 ± 10.4 cm/s, pressure gradient was 1.8 ± 0.4 mmHg, pulsatility index was 1.4 ± 0.3, resistance index was 0.9 ± 0.02, VTI was 25.02 ± 6.2 cm, systolic/diastolic was 19 ± 4.7 and heart rate was 95.7 ± 28.9 per minute. For Myocardial Performance Index (LV)–Tei Index, the mean ejection, isovolumic relaxation, isovolumic contraction time and myocardial performance index were 0.24 ± 0.01, 0.14 ± 0.01, 0.14 ± 0.02 and 1.2 ± 0.1 s, respectively. The results of the present study provide the reference values of PW echocardiographic parameter measurements in normal adult donkeys. Such reference values are helpful, especially when confronted with clinical cases with cardiovascular disorders.

## 1. Introduction

Nowadays, donkeys are of high interest, both for clinicians and owners, due to their inner working abilities and as companion animals. Thus, in recent years, people’s awareness of the well-being and care of these animals has continued to increase, and the demand for professional veterinary services has increased. According to reports, there are many differences between donkeys and horses. Therefore, clinical data, treatment and diagnostic protocols from horse to donkey may lead to misdiagnosis and unnecessary or inadequate treatment [1]. It was found that cardiovascular disease is underreported in donkeys, possibly related to their limited athletic posture and frequent poor performance-related examinations. Reports on treatments for cardiovascular disease are anecdotal in donkeys [2]. Normal echocardiographic parameters have also been reported in healthy donkeys [3]. Amory et al. [4] described normal values of echocardiographic dimensions and functional indexes, as well as quantitative reference values for Doppler flow in healthy donkeys. However, those references must be taken with restraint because marked variations can be seen depending on the breed, body size, age, growth rate and training of the animal.

Pulsed wave (PW) Doppler uses short ultrasound bursts, which are transmitted to a point (designated as “sample volume”) distant from the transducer [5]. PW Doppler allows the calculation of blood flow velocity, direction and spectral characteristics from a specified point in the heart or blood vessel, but the measurement of the maximum velocity is limited as the pulse repetition frequency is limited [6]. Consequently, it is used for the evaluation of hemodynamic abnormalities in the heart, myocardial and pericardial disorders, transvalvular gradients, intracardiac pressures and shunts, diastolic and systolic cardiac performance and severity of valvular lesions. In humans, assessment of ventricular function can distinguish patients with normal LV function and LV dysfunction and provide knowledge about the hemodynamic alterations as a side effect of the use of therapeutic agents [7,8]. Blood flow velocity patterns are altered in human patients with cardiac dysfunction and valvular regurgitation [9]. Mitral and aortic valve regurgitation can be assessed by regurgitant volume or volumetric volume [10,11].

In humans and horses, the velocity time integral (VTI) is a hemodynamic echocardiographic parameter measured from the Doppler spectrum across the valves through the left ventricular outflow tract (LVOT) [12]. The area under the flow velocity curve represents the distance of the blood volume passing through the valve [13]. Doppler echocardiography has been used in the assessment of valvular regurgitation in horses with cardiac murmurs [14,15,16]. Moreover, peak blood flow velocity in the mitral valve has been assessed in healthy warm blood horses using PW Doppler echocardiography [17]. Furthermore, Blissitt and Bonagura [18] measured peak velocity and deceleration time of mitral inflow E-wave and peak velocity, peak acceleration, acceleration time and VTI of aortic outflow using Doppler echocardiography in thoroughbred and thoroughbred cross horses.

In horses, the pressure gradient, which was calculated using a simplified Bernoulli equation, is applied to insufficient valves, ventricular and atrial septal defects, and intracardiac pressure determination [19]. The pressure half time (PHT) is the time interval in milliseconds between the maximal mitral gradient in early diastole and the time point where the gradient is half the maximum initial value [20]. It is a simple Doppler method used for assessing the MVA, severity of aortic regurgitation and pressure deceleration [21].

Till now, echocardiographic parameters, including pulsatility index (PI), resistance index (RI) and myocardial performance index (MPI), which are used in human studies to evaluate the cardiac performance, was not recorded in horses with cardiac disorders [22,23]. PI and RI are useful in the measurement of blood flow resistance [24]. The myocardial performance index is an easily measured index used for the assessment of global heart function, combining both systolic and diastolic cardiac performance [22].

Data regarding reference values of Doppler echocardiographic parameters in healthy donkeys are scarce. Consequently, the current study was conducted to determine the reference values and repeatability for PW Doppler echocardiographic variables of the mitral valve, aortic valve and myocardial performance in normal donkeys.

## 2. Materials and Methods

### 2.1. Experimental Animals

Thirty clinically healthy donkeys (*Equus asinus*) were used for this study. The age of the donkeys was 5 to 9 years (7.2 ± 1.43), and their body weight was 100 to 220 Kg (172 ± 40.49). None of the donkeys had cardiovascular disorders nor any evidence of other systemic diseases based on clinical examination. All donkeys were free from significant valvular regurgitation using Doppler echocardiography. All donkeys under investigation were housed in straw-bedded boxes in the animal house of the veterinary teaching hospital, Faculty of Veterinary Medicine, Mansoura University. All donkeys were fed twice per day with 1.5 kg hay/100 kg B.W. and 1.5 kg concentrate with ad libitum water access at least two weeks before the trials. This study was carried out at the Department of Internal Medicine and Infectious Diseases, Faculty of Veterinary Medicine, Mansoura University, Mansoura, Egypt. The study was approved by the Animal welfare and Ethics Committee, Faculty of Veterinary Medicine, Mansoura University (approval no.R-136-2021).

### 2.2. Echocardiographic Examination

Transcutaneous echocardiographic examinations were performed according to the standard methods described by Youssef et al. [25]. All echocardiographic procedures and precautions were followed according to the recommendations of the American Society of Echocardiography. The 2-dimensional (2-D), Colour flow mapping and spectral Doppler modes were performed with a CHISON Digital Color Doppler Ultrasound System, iVis 60 EXPERT VET, (CHISON Medical Imaging Co., Ltd., Wuxi, China), using a 2–3.9 MHz phased array transducer, with a maximal depth of 24.1 cm. During spectral Doppler recording, the transducer was used in the high pulsed repetition frequency mode (HPRF) using a frequency of 6 MHz. The velocity scale was set at 150 cm/s so that only one sample volume was available for velocity recording. Flow velocities and time were displayed graphically, and accurate velocity recordings were obtained when the Doppler ultrasound beam was aligned parallel to the direction of flow according to [26]. Alignment with blood flow was initially assessed using a 2-D ultrasound image. A color flow study was used as a guide to determine the place of the sampling site in an area of maximal blood velocity. PW Doppler measurements were conducted according to guidelines previously described in [18,27]. For mitral inflow, a left parasternal long axis apical view of the left ventricular inlet was used [27]. For aortic outflow, a left parasternal long axis view (5-chambered) of the left ventricular outflow tract (LVOT) was used [27]. For a myocardial performance index of the left ventricle (MPILV), the apical five-chamber view was used [28].

### 2.3. PW Doppler Measurements

For mitral inflow, the sample volume was placed on the ventricular side of the mitral valve at the valve tips, with minor adjustments in transducer angles to obtain the flow velocity (Figure 1a,b). The velocity was measured during the rapid filling phase of the ventricle (E wave) and during the atrial contraction (A wave). Based on those, the duration of the E-wave and A-wave can be calculated. The pressure gradient was described according to the method described by Weyman [29]. Mitral PHT was calculated using the deceleration time, which is known as the time from the peak mitral velocity on the velocity decline extrapolated to the baseline [20]. The MVA was measured from PHT using an empirical formula as described in [30].

For aortic outflow, the sample volume was placed on the arterial side of the aortic valve [27] to obtain the velocity of blood flow. The pressure gradient was described according to Weyman 1994 [29].

PI and RI for mitral and aortic valves can be calculated according to standard methods [31,32]. The VTI under the velocity waveform was measured by tracing the modal velocity, which was represented by the bright line in the spectral Doppler waveform envelopes from the Doppler signal [33].

For the myocardial performance index of the left ventricle (MPILV), the sample volume was placed at the tips of the mitral valve leaflets. The isovolumic contraction time (ICT) interval was measured from the end to the onset of mitral inflow. Meanwhile, the isovolumic relaxation time (IRT) interval was measured at the time between the onset and the end of LV outflow when the sample volume was placed below the aortic valve. The myocardial performance index was later calculated from the equation (ICT + IRT)/ET [28].

### 2.4. Repeatability

Each donkey was examined via echocardiography three times at one-week intervals by the same observer. Each day, 3 PW echocardiographic measurements with a variance <5% were recorded on 3 non-consecutive cardiac cycles. Measurements with a variance >5% were discarded. Thus, by the end of the study, there were 9 PW echocardiographic measurements for each donkey.

### 2.5. Statistical Analysis

Statistical analysis followed the American Society for Veterinary Clinical Pathology (ASVCP) reference interval guidelines. Specifically, a histogram was used to illustrate the reference values. The data of PW echocardiography were checked for normality by the Kolmogorov–Smirnov normality test. The result revealed normally distributed data. The summary statistics and the frequency distribution (mean ± SD, 95% CI, median, range, and 10th, 25th, 75th, and 90th percentiles) of the PW echocardiographic measurements for the mitral and aortic valve in all donkeys were reported. The reproducibility of the PW echocardiographic measurements was assessed by calculation of the intra-assay and interassay coefficient of variation (CVs). For each of the 3 days that data were collected, the intra-assay CV was calculated by dividing the SD of the measurements for that day by the mean of the measurements for that day. For each donkey, the interassay CV was calculated for each of the 3 measurements (1, 2, and 3) of PW echocardiographic measurements obtained on each data collection day by dividing the SD for that particular measurement by the mean for that particular measurement. To assess the variation in the intra-assay CV among days and the interassay CV% among reads, repeated measures ANOVA was performed. Mauchly’s sphericity test was used to detect the significant variations. When there was a significant result, one-way ANOVA with post-hoc Duncan multiple comparison tests were used to detect the specific variations. For all analyses, values of *p* < 0.05 were considered significant. All statistical procedures were performed with a commercially available software program (Graph Pad prism, San Diego, CA, USA).

## 3. Results

Frequency distribution for echocardiographic parameters of the mitral valve, aortic valve and myocardial Performance Index (LV)—Tei Index evaluated by PW Doppler echocardiography in healthy donkeys (*Equus asinus*) were summarized (Table 1, Table 2 and Table 3).

For echocardiographic parameters of the mitral valve, aortic valve and myocardial Performance Index (LV)—Tei Index evaluated by PW Doppler echocardiography in healthy donkeys (*Equus asinus*) were summarized (Table 4, Table 5 and Table 6).

The intra-assay and interassay coefficient of variation (CVs) of the mitral valve in healthy donkeys using PW Doppler echocardiography were summarized (Table 7 and Table 8).

The intra-assay and interassay CVs of the aortic valve in healthy donkeys using PW Doppler echocardiography were summarized (Table 9 and Table 10).

The intra-assay and interassay CVs of myocardial Performance Index (LV)–Tei Index in healthy donkeys using PW Doppler echocardiography were summarized (Table 11 and Table 12).

## 4. Discussion

PW Doppler is used in combination with the 2D image to assess flow velocities within discrete regions of the heart and great vessels, which are used to evaluate the cardiac performance [21]. In the current study, E-wave was higher than A-wave for the mitral inflow during filling of the left ventricle in healthy donkeys. The same results were recorded in normal thoroughbred and thoroughbred cross horses [18], normal dogs [34] and normal human subjects [35].

PW Doppler evaluates the mitral velocity, which provides intuition into the dynamics of LV filling and helps to evaluate the diastolic function [36]. Furthermore, the evaluation of transmitral velocity, together with tricuspid and hepatic vein velocities, is useful when evaluating cardiac tamponade and constrictive pericarditis [37].

The ventricular filling results from isovolumic relaxation, ventricular compliance, filling pressures from the left atrium to left ventricle, pericardial restraint, ventricular interaction and atrial function, and it may be influenced by afterload and contractility [7]. Consequently, changes in the ventricular relaxation affect early filling of the left ventricular chamber, while changes in ventricular compliance affect late diastolic filling of the ventricle with a resultant increase of E-wave [38]. Moreover, E-wave velocity is increased with increased left atrial pressure, decreased left ventricular pressure associating the increased rate of ventricular relaxation, and decreased MVA [39]. Meanwhile, the early ventricular filling is decreased in cases of decreased atrial pressure, decreased rate of ventricular relaxation, increased ventricular compliance and increased MVA with a subsequent decrease of transmitral E-wave amplitude, which resulted in increased A-wave velocity as late diastole contributes more to total left ventricular filling [39]. Thus, changes in the velocity occur with alterations in the left atrial and left ventricular diastolic pressures [40].

The pressure gradient of the E-wave is higher than the A-wave of the mitral inflow during filling of the left ventricle in healthy donkeys. In equine, the pressure gradient is a reflection of normal intra-cardiac pressure and pathological increased pressure [41]. The pressure gradient quantifies the severity of stenotic lesions and can differentiate the unknown pressure from the known pressure. The pressure gradient will be increased in the presence of conduct-type lesions as tunnel sub-valvular stenosis and in cases of decreased blood viscosity. In contrast, the increased blood viscosity may underestimate the pressure gradient [42].

In the current study, the velocity of the aortic outflow is 64.9 ± 10.4 during the first third of systole in healthy donkeys. The aortic flow pattern is found in normal thoroughbred and thoroughbred cross horses (0.937 ± 0.094) [18], in normal adult Turkmen horses (101.948 ± 15.341) [43], in clinically normal dogs (106.0 ± 21.0) [34] and in human (92 ± 11) [44]. This is probably due to differences in alignment with aortic flow in donkeys. However, this may represent a species variation in actual flow velocities.

The flow velocity of aortic flow is affected by heart rate. A faster heart rate will increase peak and mean velocity [42]. The velocity of blood flow depends on the blood volume moving through the vessels or orifice. When there is a high blood flow [18], severe valvular insufficiency or stenosis, a coexisting shunt, anemia and/or sepsis [45], the pressure gradient will be inaccurate.

In the current study, the MVA equals 2.4 ± 1.5 cm^2^, and the PHT equals 135 ± 93.9 ms in healthy donkeys. However, in humans, the MVA is found to be 4.0–6.0, and the PHT is 40–70 ms [20].

The Doppler study can be used to calculate pressure half-time (PHT), which is defined as the time required for the pressure gradient across an obstruction to decrease to half of its maximal value. Thus, PHT increases as the severity of stenosis increases.

Overestimation of the MVA occurs when PHT is shortened by concomitant significant aortic insufficiency, decreased ventricular compliance and atrial septal defect [21]. PHT is useful in the detection of mitral stenosis with coexistent mitral regurgitation [21].

In patients with aortic regurgitation, the Doppler velocity becomes significantly shorter <250 ms because of the rapid increase in left ventricular diastolic pressure and decrease in aortic pressure. Furthermore, it may be affected by severe diastolic dysfunction with marked elevation of left ventricular diastolic pressure without severe aortic regurgitation [30].

In the current study, PI was 1.4 ± 0.4, RI was 0.9 ± 0.03 for the mitral valve, the PI was 1.4 ± 0.3 and RI was 0.9 ± 0.02 for the aortic valve in healthy donkeys. Normal values for the PI and RI are 1.36–1.56 and 0.6–0.8, respectively.

PI is equal to the difference between the peak systolic velocity and the minimum diastolic velocity divided by the mean velocity during the cardiac cycle. The value of PI decreases with distance from the heart [46]. The two indices, pulsatility index and resistance index, measure the resistance of blood flow. Furthermore, they are affected by input pressure waveform pulsatility, impedance and resistance changes [24].

In the current study, the VTI is 19.1 ± 5.7 cm for mitral inflow and equals 25.02 ± 6.2 for aortic outflow in healthy donkeys. The VTI is found to be 25.369 ± 3.209 cm in normal horses for aortic flow [18], 0.146 ± 0.029 cm in dog for aortic outflow [34] and 25.1 ± 3.4 cm in humans for aortic flow [47]. Left ventricular outflow tract velocity time integral (LVOT VTI) is a measure of cardiac systolic function and cardiac output. Heart failure patients with low cardiac output are known to have poor cardiovascular outcomes. Thus, extremely low LVOT VTI may predict heart failure patients at highest risk for mortality [12].

In the present study, Myocardial Performance Index (LV)–Tei Index was 1.7 ± 0.7, isovolumic contraction time was 0.3 ± 0.1, isovolumic relaxation time was 0.3 ± 0.1, and ejection time was 0.4 ± 0.1. The MPI was found to be 0.52 ± 0.12 in dog [48] and in human (0.39 ± 0.05) [22]. Systolic dysfunction prolongs pre-ejection (ICT) and a shortening of the ejection time (ET). Both systolic and diastolic dysfunction results in abnormality in myocardial relaxation, which prolongs the relaxation period (IRT) [22].

The Myocardial Performance Index (LV)–Tei Index is independent of heart rate, arterial blood pressure, ventricular geometry, atrioventricular valve regurgitation, loading condition as afterload and preload, and can be used to evaluate the function of both the LV and the RV [49]. The Myocardial Performance Index (LV)–Tei Index have strong prognostic value in severe cardiac diseases such as cardiac amyloidosis (0.54 ± 0.16), dilated cardiomyopathy (0.59 ± 0.10), ischemic heart diseases (0.85 ± 0.32), pulmonary hypertension (0.93 ± 0.34), congestive heart failure (0.37 ± 0.05), valvular diseases (0.6 ± 0.2), myocardial infarction, congenital heart diseases (0.35 ± 0.03) and cardiotoxicity (0.45 ± 0.06) [50].

## 5. Conclusions

Results of the present study provided an initial reference for PW (PW) Doppler echocardiographic variables of the mitral valve, aortic valve and myocardial performance in donkeys, which will be beneficial for clinicians who perform cardiac examinations in these animals. The intra-assay and interassay CVs for the mitral valve, aortic valve measurements and myocardial performance indicated that the technique is a feasible and precise method for determining cardiac measurements and functions in donkeys.

## Figures and Tables

**Figure 1 animals-12-02296-f001:**
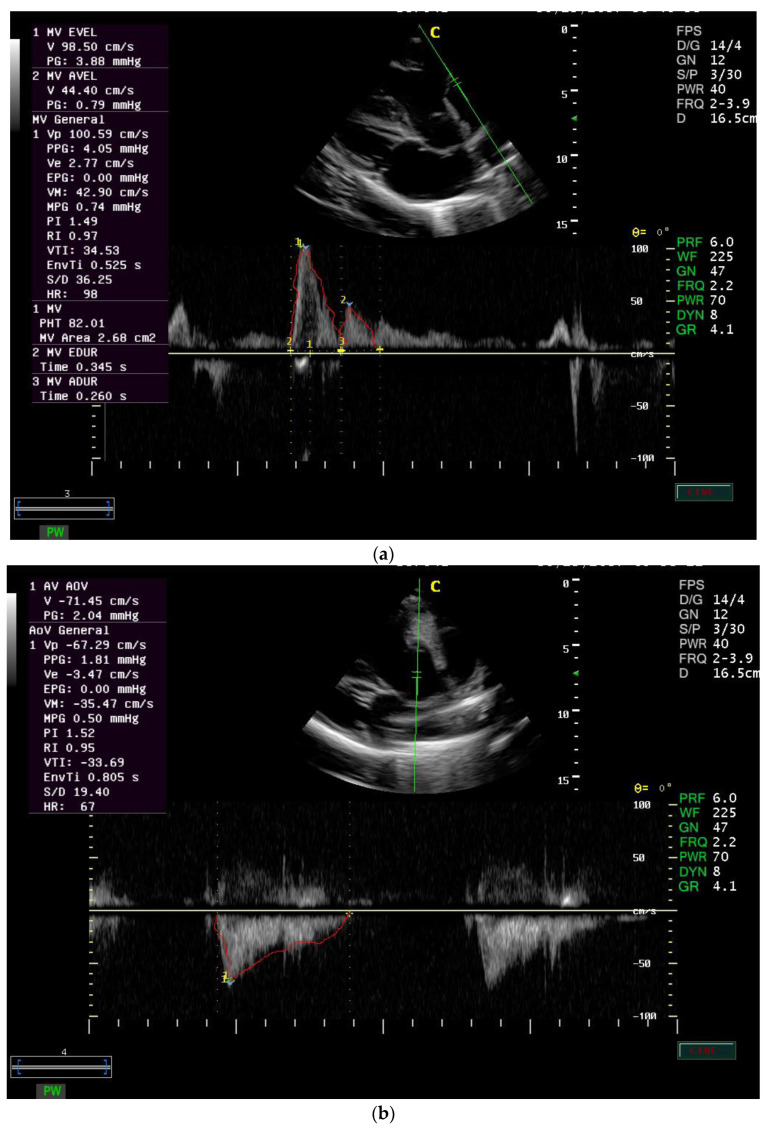
(**a**) Left parasternal long axis view of the left ventricular inlet of mitral valve view in one of the 30 normal adult donkeys. (**b**) Left parasternal long axis view (5-chambered) of left ventricular outflow tract (LVOT) with the sample volume on the arterial side of the aortic valve in 30 normal adult donkeys.

**Table 1 animals-12-02296-t001:** Frequency distribution of pulsed wave doppler echocardiographic measurements of mitral valve in normal donkeys.

Variables	Highest Frequency		Lowest Frequency	
	Value	No (270) (%)	Value	No (270) (%)
V E wave (cm/s)	≥12–14	66 (24.4%)	≥42–46	3 (1.1%)
PG E wave (mmHg)	≥15–25	63 (23.3%)	≥85–95	3 (1.1%)
Dur E wave (s)	≥15–25	62 (22.9%)	≥60–75	12 (4.4%)
V A wave (cm/s)	≥22–26	51 (18.9%)	≥40–46	3 (1.1%)
PG A wave (mmHg)	≥50–60	48 (17.8%)	≥80–90	3 (1.1%)
Dur A wave (s)	≥15–25	69 (25.6%)	≥60–70	9 (3.3%)
MV area (cm^2^)	≥30–50	72 (26.7%)	≥100–120	3 (1.1%)
PHT (ms)	≥30–50	72 (26.7%)	≥100–120	9 (3.3%)
PI	≥15–25	75 (27.8%)	≥80–95	6 (2.2%)
RI	≥1.5–2.5	77 (28.5%)	≥3–4	7 (2.5%)
VTI (cm)	≥10–14	39 (14.4%)	≥34–38	9 (3.3%)
S/D	≥10–25	66 (24.4%)	≥60–70	3 (1.1%)

V E wave: Velocity E wave; PG E wave: pressure gradient E wave; Dur E wave: duration E wave; VA wave: Velocity A wave; PG A wave: pressure gradient A wave; Dur A wave: duration A wave; MV area: mitral valve area; PHT: pressure half time; PI: pulsatility index; RI: resistance index; VTI: velocity time integral; S/D: systolic/diastolic.

**Table 2 animals-12-02296-t002:** Frequency distribution of PW doppler echocardiographic measurements of aortic valve in normal donkeys.

Variables	Highest Frequency		Lowest Frequency	
	Value	No (270) (%)	Value	No (270) (%)
V (cm/s)	≥12–16	60 (22.2%)	≥15–17	6 (2.2%)
PG (mmHg)	≥15–25	88 (32.5%)	≥60–65	7 (2.5%)
PI	≥15–25	72 (26.6%)	≥50–60	3 (1.1%)
RI	≥0.5–1.5	99 (36.6%)	≥5.5–6	3 (1.1%)
VTI (cm)	≥15–25	59 (21.8%)	≥55–65	3 (1.1%)
S/D	≥0–20	72 (26.6%)	≥90–100	3 (1.1%)

V: Velocity; PG: pressure gradient; PI: pulsatility index; RI: resistance index; VTI: velocity time integral; S/D: systolic/diastolic.

**Table 3 animals-12-02296-t003:** Frequency distribution of PW doppler echocardiographic measurements of myocardial performance index (LV)—Tei Index valve in normal donkeys.

Variables	Highest Frequency		Lowest Frequency	
	Value	No (270) (%)	Value	No (270) (%)
ET (s)	≥15–25	75 (27.8%)	≥45–50	9 (3.3%)
IRT (s)	≥25–35	57 (21.1%)	≥70–85	15 (5.5%)
ICT (s)	≥30–40	78 (28.9%)	≥0–10	3 (1.1%)
MPI	≥6–10	63 (23.3%)	≥24–28	12 (4.4%)

ET: ejection time; IRT: isovolumic relaxation time; ICT: isovolumic contraction time; MPI: myocardial performance index.

**Table 4 animals-12-02296-t004:** Summary statistics of PW doppler echocardiographic measurements of mitral valve in normal donkeys.

Variables	Mean ± SD	95% CI	Median (Range)	Percentile			
				10%	25%	75%	90%
V E wave (cm/s)	57.7 ± 12.4	56.2–59.2	53.4 (40.9–98.5)	45.1	50.6	62.4	76.7
PG E wave (mmHg)	1.4 ± 0.7	1.3–1.5	1.1 (0.7–3.9)	0.8	1.03	1.6	2.4
Dur E wave (s)	0.4 ± 0.14	0.3–0.4	0.4 (0.2–0.9)	0.2	0.3	0.4	0.5
V A wave (cm/s)	32.3 ± 9.1	31.2–33.4	31.6 (16.7–61.7)	22.2	24.9	37.5	42.3
PG A wave (mmHg)	0.5 ± 0.3	0.4–0.5	0.4 (0.1–1.5)	0.2	0.3	0.6	0.7
Dur A wave (s)	0.3 ± 0.1	0.2–0.3	0.3 (0.1–0.6)	0.1	0.2	0.3	0.4
MV area (cm^2^)	2.4 ± 1.5	2.2–2.6	1.9 (0.5–7.9)	0.7	1.4	2.9	4.3
PHT (ms)	91.3 ± 24.9	88.3–94.2	102.6 (27.6–125.3)	51.3	75.7	109.8	116.1
1.4 (0.01–2)	1.3–1.4	1.4 ± 0.4	PI		1.2	1.6	1.8
0.96 (0.8–1.1)	0.94–0.96	0.9 ± 0.03	RI		0.9	1	1.1
18.8 (9.9–36.5)	18.5–19.8	19.1 ± 5.7	VTI (cm)		15.1	22.4	26.5
22.5 (9.1–37.5)	21.9–23.7	22.8 ± 7.5	S/D		16	28.5	33.9

V E wave: Velocity E wave; PG E wave: pressure gradient E wave; Dur E wave: duration E wave; VA wave: Velocity A wave; PG A wave: pressure gradient A wave; Dur A wave: duration A wave; MV area: mitral valve area; PHT: pressure half time; PI: pulsatility index; RI: resistance index; VTI: velocity time integral; S/D: systolic/diastolic.

**Table 5 animals-12-02296-t005:** Summary statistics of PW Doppler echocardiographic measurements of aortic valve in normal donkeys.

Variables	Mean ± SD	95% CI	Median (Range)	Percentile			
				10%	25%	75%	90%
V (cm/s)	64.9 ± 10.4	63.8–66.2	64.9 (42.3–86.7)	49.4	58.6	71.6	79.7
PG (mmHg)	1.7 ± 0.5	1.7–1.8	1.7 (0.7–3)	0.9	1.4	2.1	2.5
PI	1.4 ± 0.3	1.3–1.4	1.4 (0.5–1.9)	0.96	1.2	1.6	1.7
RI	0.9 ± 0.02	0.95–0.96	0.95 (0.9–1.02)	0.92	0.94	0.96	0.98
VTI (cm)	25.02 ± 6.2	24.3–25.8	26.3 (11.8–38.02)	15.7	20.3	29.9	32.7
S/D	23.6 ± 12.04	22.2–25.1	21.3 (6.7–86)	13.2	17.3	25.5	36

V: Velocity; PG: pressure gradient; PI: pulsatility index; RI: resistance index; VTI: velocity time integral; S/D: systolic/diastolic.

**Table 6 animals-12-02296-t006:** Summary statistics of PW Doppler echocardiographic measurements of myocardial performance index (LV)—Tei Index in normal donkeys.

Variables	Mean ± SD	95% CI	Median (Range)	Percentile			
				10%	25%	75%	90%
ET (s)	0.4 ± 0.1	0.4 ± 0.1	0.4 (0.2–0.6)	0.2	0.3	0.4	0.5
IRT (s)	0.3 ± 0.2	0.3 ± 0.2	0.3 (0.1–0.8)	0.1	0.2	0.4	0.5
ICT (s)	0.3 ± 0.1	0.3 ± 0.1	0.3 (0.1–0.7)	0.1	0.2	0.4	0.5
MPI	1.7 ± 0.7	1.7 ± 0.7	1.7 (0.6–3.3)	0.9	1.3	2.1	2.9

ET: ejection time; IRT: isovolumic relaxation time; ICT: isovolumic contraction time; MPI: myocardial performance index.

**Table 7 animals-12-02296-t007:** Intra-assay CVs of pulsed wave doppler echocardiographic measurements of mitral valve in normal donkeys.

Variables	Mean ± SD	95% CI	Median (Range)	Percentile			
				10%	25%	75%	90%
V E wave (cm/s)	13.01 ± 9.5	11.1–14.9	12.6 (0.8–45.8)	2.8	4.9	16.4	22.9
PG E wave (mmHg)	25.9 ± 18.9	21.9–29.8	24.3 (1.9–90.7)	5.6	9.9	32.2	47.2
Dur E wave (s)	24.8 ± 17.2	21.1–28.4	20.02 (0.8–71.4)	3.1	13.6	31.4	50.8
V A wave (cm/s)	20.9 ± 8.3	19.2–22.7	21.2 (5.1–35.2)	7.7	14.9	25.7	31.1
PG A wave (mmHg)	40.6 ± 15.9	37.3–43.9	41.9 (9.5–69.1)	14.6	28.4	52.8	62.5
Dur A wave (s)	27.8 ± 15.5	24.6–31.1	23.8 (3.7–69.6)	10.3	16.5	40.7	44.8
MV area (cm^2^)	46.3 ± 25.3	41.1–51.6	43.8 (9.4–119.6)	17.7	26.4	57.8	87.9
PHT (ms)	49. 1 ± 24.2	43.9–54.1	47.1 (9.9–91.4)	18.1	25.7	68.3	84.5
PI	23.1 ± 19.1	19.1–27.1	15.3 (3.2–85.6)	6.7	12.1	28.1	50.2
RI	1.8 ± 1.5	1.4–2.1	1.5 (0.1- 6.6)	0.6	0.61	2.2	5.3
VTI (cm)	21.2 ± 11.2	18.9–23.6	19.6 (5.7–45.4)	7.2	11.6	25.3	42.5
S/D	32.8 ± 19.5	28.8–36.9	28.6 (2.2–89.8)	14.9	20.9	37.3	62.1

V E wave: Velocity E wave; PG E wave: pressure gradient E wave; Dur E wave: duration E wave; VA wave: Velocity A wave; PG A wave: pressure gradient A wave; Dur A wave: duration A wave; MV area: mitral valve area; PHT: pressure half time; PI: pulsatility index; RI: resistance index; VTI: velocity time integral; S/D: systolic/diastolic.

**Table 8 animals-12-02296-t008:** Interassay CVs of PW doppler echocardiographic measurements of mitral valve in normal donkeys.

Variables	Mean ± SD	95% CI	Median (Range)	Percentile			
				10%	25%	75%	90%
V E wave (cm/s)	13.5 ± 7.1	11.9–14.9	13.1 (1.5–30.1)	3.2	7.9	16.8	22.7
PG E wave (mmHg)	26.5 ± 13.5	23.7–29.3	26.7 (3.2–53.9)	6.2	15.8	34.2	46.1
Dur E wave (s)	27.6 ± 17.6	23.9–31.3	24.6 (5.7–67.1)	6.7	12.9	37.7	57.3
V A wave (cm/s)	19.9 ± 9.6	17.9–21.9	20.9 (4.5–44.8)	5.8	13.2	26.4	31.2
PG A wave (mmHg)	38.7 ± 18.5	34.8–42.5	42.3 (9.2–87.9)	11.5	26.2	48.8	60.9
Dur A wave (s)	28.3 ± 15.3	25.1–31.5	21.8 (11.4–65.2)	12.9	18.3	37.6	56.4
MV area (cm^2^)	48.1 ± 26.4	42.5–53.6	39.4 (6.9–93.5)	15.1	26.4	68.5	91.02
PHT (ms)	51.4 ± 30.8	44.9–57.8	44.8 (6.9–110.7)	13.9	27.6	79.9	97.8
PI	22.7 ± 17.8	18.9–26.4	21.2 (4.7–92.6)	6.5	11.1	27.4	33.8
RI	2.3 ± 1.8	1.8–2.1	1.7 (0.6–6.9)	0.6	1.1	2.6	6.5
VTI (cm)	21.9 ± 8.5	20.2–23.7	23.8 (6.9–34.9)	11.3	15.3	28.5	31.7
S/D	36.1 ± 20.6	31.7–40.4	34.1 (5.8–91.9)	11.1	23.9	41.1	76.4

V E wave: Velocity E wave; PG E wave: pressure gradient E wave; Dur E wave: duration E wave; VA wave: Velocity A wave; PG A wave: pressure gradient A wave; Dur A wave: duration A wave; MV area: mitral valve area; PHT: pressure half time; PI: pulsatility index; RI: resistance index; VTI: velocity time integral; S/D: systolic/diastolic.

**Table 9 animals-12-02296-t009:** Intra-assay CVs of PW doppler echocardiographic measurements of aortic valve in normal donkeys.

Variables	Mean ± SD	95% CI	Median (Range)	Percentile			
				10%	25%	75%	90%
V (cm/s)	11.6 ± 6.9	10.2–13.1	10.4 (1.3–29.7)	2.6	5.6	15.7	20.9
PG (mmHg)	23.04 ± 13.6	20.6–25.5	20.9 (2.7–59.3)	5.3	13.7	30.2	39.9
PI	17.1 ± 11.5	14.7–19.5	14.9 (3.4–45.2)	3.9	7.2	25.2	37.2
RI	1.9 ± 1.3	1.7–2.3	1.6 (0.6–5.4)	0.6	1.1	3.1	3.9
VTI (cm)	19.2 ± 13.1	16.4–21.9	15.1 (1.2–60.4)	4.7	10.2	30.5	34.1
S/D	32.6 ± 21.6	28.1–37.1	31.2 (5.5–81.8)	7.4	15.2	40.3	66.3

V: velocity; PG: pressure gradient; PI: pulsatility index; RI: resistance index; VTI: velocity time integral; S/D: systolic/diastolic.

**Table 10 animals-12-02296-t010:** Interassay CVs of PW doppler echocardiographic measurements of aortic valve in normal donkeys.

Variables	Mean ± SD	95% CI	Median (Range)	Percentile			
				10%	25%	75%	90%
V (cm/s)	13.6 ± 6.9	12.2–15.1	13.7 (0.9–31.4)	4.9	8.3	17.7	24.1
PG (mmHg)	26.9 ± 13.3	24.2–29.8	26.7 (2.04–63.9)	9.7	17.4	33.5	46.9
PI	20.6 ± 11.6	18.3–23.1	18.5 (3.6–59.7)	6.1	13.8	28.6	35.3
RI	2.03 ± 1.3	1.7–2.3	1.6 (0.6–6.5)	0.6	1.1	2.5	3.9
VTI (cm)	22.9 ± 13.1	20.2–25.7	18.6 (6.9–47.9)	7.6	13.3	32.6	43.9
S/D	35.6 ± 22.1	31.1–40.2	31.2 (7.6–93.1)	12.1	19.4	51.5	72.2

V: velocity; PG: pressure gradient; PI: pulsatility index; RI: resistance index; VTI: velocity time integral; S/D: systolic/diastolic.

**Table 11 animals-12-02296-t011:** Intra-assay CVs of PW doppler echocardiographic measurements of myocardial performance index (LV)—Tei Index in normal donkeys.

Variables	Mean ± SD	95% CI	Median (Range)	Percentile			
				10%	25%	75%	90%
ET (s)	19.5 ± 8.4	17.7–21.2	19.4 (4.8–34.5)	9.3	10.8	23.8	33.6
IRT (s)	44.3 ± 19.5	40.3–48.4	40.4 (24.5–80.9)	24.8	27.8	56.8	80.1
ICT (s)	38.7 ± 11.3	36.3–40.9	37.7 (25.01–63.9)	25.9	29.1	43.8	56.7
MPI	30.2 ± 14.7	33.2–37.6	33.2 (5.92–54.4)	13.6	18.6	40.8	53.9

ET: ejection time; IRT: isovolumic relaxation time; ICT: isovolumic contraction time; MPI: myocardial performance index.

**Table 12 animals-12-02296-t012:** Interassay CVs of PW doppler echocardiographic measurements of myocardial performance index (LV)–Tei Index in normal donkeys.

Variables	Mean ± SD	95% CI	Median (Range)	Percentile			
				10%	25%	75%	90%
ET (s)	28.03 ± 12.8	25.4–30.7	28.2 (5.3–49.8)	8.2	21.1	39.4	42.9
IRT (s)	38.7 ± 17.6	35.01–42.4	38.3 (10.9–66.8)	14.9	20.3	53.8	63.4
ICT (s)	37.4 ± 17.3	33.9–42.9	36.7 (3.5–71.1)	14.5	22.1	47.8	65.5
MPI	36.4± 15.6	35.6–46.2	38.7 (9.2–66.01)	12.3	24.5	48.6	57.03

ET: ejection time; IRT: isovolumic relaxation time; ICT: isovolumic contraction time; MPI: myocardial performance index.

## Data Availability

All data sets collected and analyzed during the current study are available from the corresponding author on fair request.

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
