# Peer review of "Reference Values and Repeatability of Pulsed Wave Doppler Echocardiography Parameters in Normal Donkeys"

_animals, 2022, doi:10.3390/ani12172296_

Round 1

Author Response

All corrections were done as recommended as track changes

Reviewer 2 Report

The manuscript is interesting and in my opinion, could be accepted after minor revision.

Please, see the detailed comments:

L 20 and 37 - PW - the abbreviation should be expended

L 22 remove the additional dot

The aim should be clearly stated in the abstract section

L 53 remove bold font

L 74 "[13]. [14-16]." if the references are correct, connect it together.

L 53 remove bold font

L 88 If the abbreviation of PI is introduced here, it should be used throughout the whole manuscript (this also applies to all other abbreviations - one is enough)

L 95-96 The aim should be stated clearly. If possible, add also your hypothesis.

L 101 remove the bold font and explain how the clinical health of donkeys was examined.

L 111 the number of permission is required

L 145 " to Weyman 1994[29]. " correct the form of citation

L 158 removes additional space

Result section - remove all bold fonts

Change  " Table (1)" to Table 1. and repeat it for the rest tables.

When the numerical data are compared in the discussion section, the units should be added

L 397 and 402 If PHT was expended above, it does not need to repeat it (line 83)

L 399 remove the underline

L 421 remove bold font

When the aim will be revised, please revise also the conclusion to be in line with new aims.

Author Response

(The authors gave the same response as above.)
